# Embedding security into ferroelectric FET array via in situ memory operation

Yixin Xu [1], Yi Xiao[1], Zijian Zhao[2], Franz Müller [3], Alptekin Vardar[3], Xiao Gong[4], Sumitha George [5], Thomas Kämpfe [3], Vijaykrishnan Narayanan[1] & Kai Ni [2] ✉

Non-volatile memories (NVMs) have the potential to reshape next-generation memory systems because of their promising properties of near-zero leakage power consumption, high density and non-volatility. However, NVMs also face critical security threats that exploit the non-volatile property. Compared to volatile memory, the capability of retaining data even after power down makes NVM more vulnerable. Existing solutions to address the security issues of NVMs are mainly based on Advanced Encryption Standard (AES), which incurs significant performance and power overhead. In this paper, we propose a lightweight memory encryption/decryption scheme by exploiting in-situ memory operations with negligible overhead. To validate the feasibility of the encryption/decryption scheme, device-level and array-level experiments are performed using ferroelectric field effect transistor (FeFET) as an example NVM without loss of generality. Besides, a comprehensive evaluation is performed on a 128 × 128 FeFET AND-type memory array in terms of area, latency, power and throughput. Compared with the AES-based scheme, our scheme shows ~22.6×/~14.1× increase in encryption/decryption throughput with negligible power penalty. Furthermore, we evaluate the performance of our scheme over the AES-based scheme when deploying different neural network workloads. Our scheme yields significant latency reduction by 90% on average for encryption and decryption processes.

The proliferation of smart edge devices has led to a massive influx of data, necessitating high-capacity and energy-efficient memory solutions for storage and processing. Traditional volatile memories, such as static random access memory (SRAM) and dynamic RAM (DRAM), are struggling to meet the demands due to their significant leakage power and low density[1]. To address this issue, high-density NVMs, such as mainstream vertical NAND flash, has become the cornerstone of modern massive information storage. NVM offers nonvolatility, zero leakage power consumption, and high density if integrated into dense 3D form[2]. Various emerging NVM technologies are being pursued targeting different levels of the memory hierarchy, e.g., as storage class memory or even as on-chip last-level cache, including 3D XPoint based on phase change memory (PCM)[3], sequential or vertical 3D resistive

memory, and back-end-of-line ferroelectric memory. Beyond simple data storage, NVM is playing an increasingly important role in data-centric computing, particularly in the compute-in-memory (CiM) paradigm. Within this paradigm, computation takes place in the analog domain within the memory array, eliminating the energy and latency associated with data transfer in conventional computing hardware. This has the potential to pave the way for sustainable data-intensive applications, particularly in the field of artificial intelligence, which is rapidly advancing with exponentially growing models. Hence NVM will be a crucial electronic component for ensuring sustainable computing in the future.

However, the nonvolatility of NVM also brings many new security challenges and concerns[4,5] that were absent in conventional volatile

[1]Pennsylvania State University, State College, PA 16802, USA. [2]University of Notre Dame, Notre Dame, IN 46556, USA. [3]Fraunhofer IPMS, Dresden, Germany. [4]National University of Singapore, Singapore, Singapore. [5]North Dakota State University, Fargo, ND 58102, USA. ✉e-mail: kni@nd.edu

memories. One of the major threats occurs when an NVM is stolen or lost, the malicious attackers may exploit the unique properties of NVM to get unauthorized accesses by low-cost tampering and then easily extract all the sensitive information stored in the devices, such as users' passwords and credit card numbers, out of the memory, and is also known as the "stolen memory attack". Compared to volatile memory such as SRAM which is considered safe due to the loss of data after power down, NVM retains data indefinitely, making them vulnerable after the system is powered down, as shown in Fig. 1d. Besides, with the increasing demand of intensive computation and the stronger desire of large data capacity, replacing some parts of storage systems with NVMs increases the incentive to attack the system and makes more data vulnerable. Hence, the security vulnerability of NVM has become a critical issue for information-sensitive systems.

To address the above issue and ensure data security in modern NVM systems, data encryption is the most common approach. AES is the most common and widely-used cryptographic algorithm[6]. It is a symmetrical block cipher algorithm including two processes—encryption and decryption, which converts the plaintext (PT) to the ciphertext (CT) and converts back by using 128-, 192-, or 256-bits keys. Because of the high security and high computation efficiency it provides, AES algorithm has attracted many researchers to actively explore its related hardware implementations and applications in a wide range of fields, such as wireless communication[7], financial transactions[8] etc. In addition, a variety of AES-based encryption techniques were proposed aiming to address the aforementioned NVM security issues and improve the security of NVM. However, AES encryption and decryption incurs significant performance and energy cost due to extra complexity involved with read and write operations, as shown in Fig. 1e. An incremental encryption scheme, called as

i-NVMM, was proposed to reduce the latency overhead[9], in which different data in NVMs is encrypted at different times depending on what data is predicted to be useful to the processor. By doing partial encryption incrementally, i-NVMM can keep the majority of memory encrypted while incurring affordable encryption overheads. However, i-NVMM relies on the dedicated AES engine that is impacted by limited bandwidth. Other prior works have proposed near-memory and in-memory encryption techniques as solutions to address the performance issues. For instance, AIM, which refers to AES in-memory implementation, supports one in-memory AES engine that provides bulk encryption of data blocks in NVMs for mobile devices[10]. In AIM, encryption is executed only when it's necessary and by leveraging the benefit of the in-memory computing architecture, AIM achieves high encryption efficiency but the bulk encryption limits support fine-grain protection. In summary, prior AES-based encryption schemes fail to efficiently address the aforementioned security issues in NVMs without incurring negligible costs. Therefore, our effort aims to break the dilemma between encryption/decryption performance and cost by finding a satisfactory solution to address the security vulnerability issue.

As illustrated in Fig. 1f, we propose a memory encryption/decryption scheme that exploits the intrinsic memory array operations without incurring complex encryption/decryption circuitry overhead. The idea is to use the intrinsic memory array operations to implement a lightweight encryption/decryption technique, i.e., bit wise XOR between the secret key and the plaintext/ciphertext, respectively. In this way, the ciphertext is written into memory through normal memory write operations and the data is secure unless a correct key, which attackers do not possess, is provided during the memory sensing operation. This work demonstrates this proposed

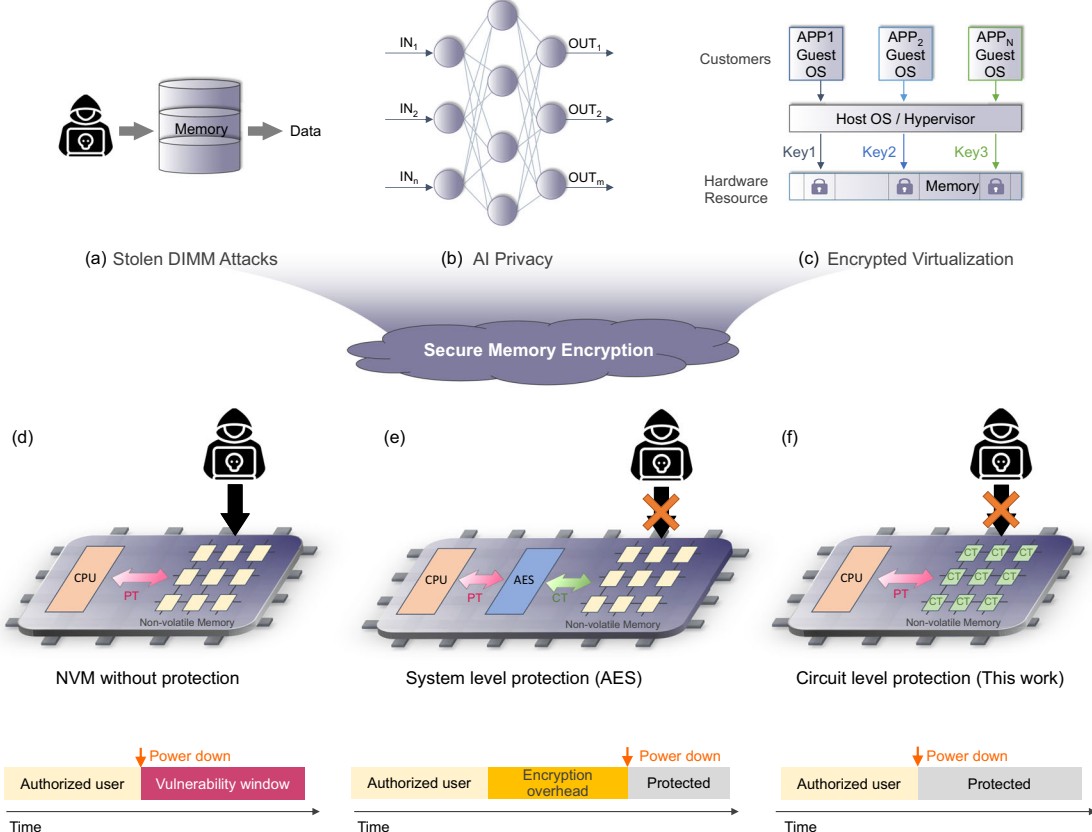

**Fig. 1 | Motivation and potential applications.** Potential applications of memory encryption techniques: (**a**) to prevent from Stolen DIMM attacks, (**b**) to ensure AI privacy, and (**c**) to implement secure encrypted virtualization (SEV), (**d**) Without protection, NVMs become vulnerable after power down. **e** NVMs with AES-

embedded can be protected after power down but with high encryption overheads. **f** With the proposed encryption scheme, NVMs can be protected after power down with minimal penalty.

encryption/decryption operation in FeFET memories and can potentially be extended to other NVM technologies. Ferroelectric $HfO_2$ has revived interests in ferroelectric memory for its scalability, CMOS compatiblity, and energy efficiency. Inserting the ferroelectric into the gate stack of a MOSFET, a FeFET is realized such that its threshold voltage ($V_{TH}$) can be programmed to the low-$V_{TH}$ (LVT) state or high-$V_{TH}$ (HVT) state by applying positive or negative write pulses on the gate, respectively. In this work, with the co-design from technology, circuit and architecture level, the proposed efficient encryption/decryption scheme can successfully remove the vulnerability window and achieve secure encryption in FeFET-based NVM. Moreover, since there is no additional complicated encryption/decryption engine (e.g. AES engine) as a part of the peripheral circuit in our architecture, our design can avoid the latency/power/area costs in AES-based encryption designs by only adding lightweight logic gates, which dramatically improves the performance of memory and expands the range of potential applications in different fields.

With the proposed memory encryption/decryption scheme integrated in FeFET memory array, many NVM-targeted attacks can be prevented. For example, if the memory device is stolen or lost, our design can effectively protect it against the malicious stolen memory attack as the attacker has no knowledge of what the data represents without correct secret keys even though they are able to physically access and read out the stored ciphertext (Fig. 1a). Besides, with negligible incurred overhead compared with normal memory, the proposed design can benefit wide applications that can exploit the added security feature without compromising performance. For instance, as shown in Fig. 1b, NVM arrays can be used to accelerate the prevalent operation in deep neural networks, i.e., matrix vector multiplication (MVM) in memory. By storing the trained neural network weights as, for example, the NVM conductance, the intended MVM operation is naturally conducted in analog domain by applying the input as input voltage pulses and summing up the resulting array column current. As artificial intelligence makes significant strides in various application domains, especially those information sensitive sectors, how to protect these trained weights from malicious entities becomes an essential problem[11,12]. Many relevant works have explored and demonstrated that data encryption embedded in CiM enables in situ authentication and computation with high area and energy efficiency[13,14]. Compared to existing AES-based encryption design which would introduce significant delay, our encryption design can efficiently encrypt and decrypt all the weights in situ and perform CiM computation with the encrypted weights directly thus ensuring high security and privacy. Another application example is secure encrypted virtualization (SEV)[15]. SEV systems require keys to isolate guests and the host OS/hypervisor from one another in order to ensure the data security in system hardware. However, present SEV systems use AES engines for encryption. By replacing the AES engines with our design, the system performance will be improved in terms of latency.

In addition, the proposed encryption strategy can work with AES together as well in order to provide higher security for some specific applications, such as SEV. For example, the AES can be adopted as the first cipher and the proposed design as the second cipher. During encryption, the plaintexts can first send to the AES engine to get the ciphertexts which would be sent as inputs of our XOR cipher to do the second encryption. The ciphertexts after these two ciphers can finally be stored in the FeFET arrays with improved security. Similarly, for decryption, the data in the memory is read out using our decryption method and then sent to AES to obtain the actual plaintexts.

## Results

### Overview of the proposed memory encryption/decryption scheme

For a deeper look into the design principles of the proposed in situ encryption/decryption scheme in FeFET array, details from different granularity and levels are demonstrated in Fig. 2. Figure 2a shows an overview of the proposed encryption memory architecture, including the FeFET-based memory array and the associated peripheral circuitry. In our encryption design, the whole memory is encrypted block-wise, which means it uses one key (1/0) per block. Depending on different cost and security demands, the granularity of encrypted blocks varies. As shown in Fig. 2b, there are three situations in the memory—unencrypted blocks, encrypted blocks with key = 1, and encrypted blocks with key = 0. For unencrypted blocks, they operate as traditional FeFET memory array. For each memory cell, depending on which data to store (1/0), FeFET would be programmed to LVT state or HVT state by applying different write voltages ($\pm V_W$). However, for encrypted blocks, each memory cell consists of two FeFETs, thus more compact than the SRAM counterpart, as illustrated in Fig. 2b. In this work, a memory array share a common body contact for high density, where a block-wise erase is performed every time a programming needs to be done. Note that bit-wise write schemes can also be adopted if single-bit programming is needed, where a column-wise body contact is adopted at the cost of memory density[16,17]. The details of the programming and inhibit schemes are discussed in the Supplementary section "Program and inhibit scheme". In addition, with different keys, these encrypted blocks follow different encryption strategies. The details of the proposed encryption/decryption strategies are demonstrated in Fig. 2c in cell level.

In the encryption process, the key is XORed with PT to obtain the CT. And the two FeFETs in the same cell would be programmed to different state patterns depending on the data that CT represents. For example, if the PT is '1' and the key for this block is '1', then the CT would be '0'. Based on our encryption strategy, the upper FeFET in the target cell should be programmed to LVT state and the bottom one should be programmed to HVT state. Similarly, if the result of CT is '1', then the upper FeFET should be set to HVT state and the bottom FeFET should be set to LVT state. In the decryption process, different read voltages ($V_R$/0 V) are applied on the gate terminals of FeFETs. However, the voltage pattern of decryption is different from that of encryption in the proposed design. The voltage pattern ($V_R$/0 or 0/$V_R$) is only relevant to the key of this cell. More specifically, if the key = 1, $V_R$ would be applied on the gate of the upper FeFET in the memory cell, and 0 V would be applied to the other FeFET. In contrast, if the key = 0, $V_R$ would be asserted on the bottom FeFET instead. In this way, original data (PT) can be successfully read out through sensing the current only when the user uses the correct key. However, for unauthorized users/attackers, even though they may have the physical access to read out the current of each memory cell, they are not aware of whether the information they read is correct or not since they don't know the correct keys for each block. Therefore, the FeFET memory are protected from information leakage and achieves intrinsic secure without extra circuit cost. Note that this design is significantly different over the SRAM based XOR encryption/decrytpion[18]. In that design, decryption is performed by reading the stored SRAM information via selectively activating the access transistor connected to BL or $\overline{BL}$, which unfortunately destroys the original symmetry of the SRAM structure, making it incompatible with normal SRAM arrays. Besides, single-ended sensing requires dedicated ADCs for both BL and $\overline{BL}$ and the CiM operation requires delicate balancing of the charging and discharging paths. None of these challenges exist for the proposed FeFET based design, making it highly appealing.

Besides, the proposed in situ memory encryption/decryption scheme is not just limited for the AND arrays. We also explore and demonstrate the feasibility of the proposed scheme to apply in other array structures, such as FeFET NAND array which provides potentially higher integration density (Supplementary Fig. S2) and FeFET NOR array (Supplementary Fig. S3). Both of them show that the proposed memory encryption/decryption scheme is general and can fit into different memory designs. Bearing the similar single

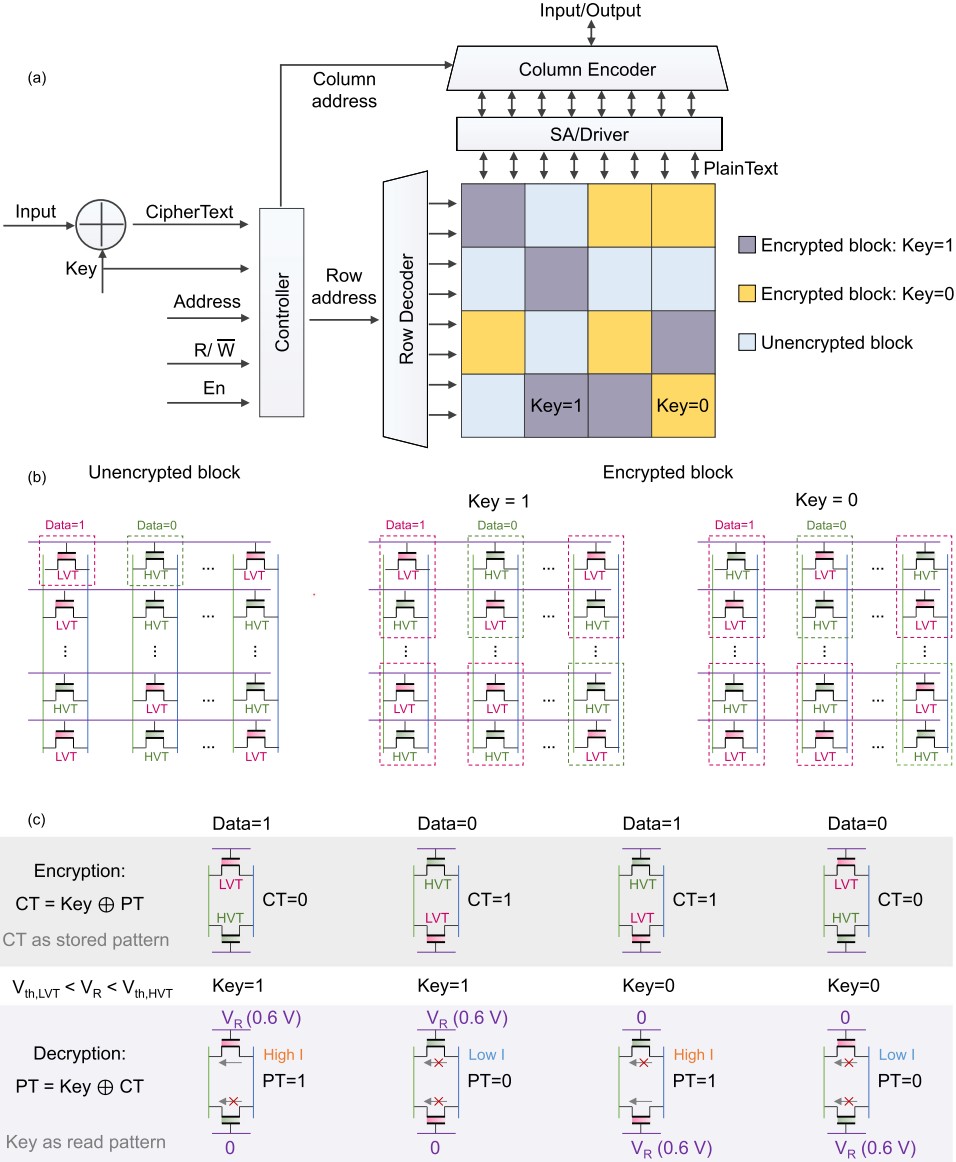

**Fig. 2 | The proposed memory encryption scheme. a** An overview of the proposed memory encryption architecture. **b** Three scenarios in the memory. **c** The details of the encryption and decryption schemes.

transistor structure, the conventional NAND and NOR flash memories can also be encrypted/decrypted with the proposed techniques. However, flash generally require a large operation voltages and a long write latency, therefore exhibiting a poor performance compared with FeFET. In both of FeFET NAND and NOR arrays, two FeFETs are coupled as one cell for representing one bit information – bit '1' or bit '0'. During the encryption process, firstly, CT will be determined by XORing PT and the corresponding key. Depending on different CT, complementary states will be programmed into the 2FeFET-based cell. During the decryption process, different read voltages depending on key patterns will be applied to the coupled FeFETs in the same cell. Finally, the correct information (PT) would be successfully read out.

## Experimental verification

In this section, functional verification of encryption/decryption operations on one single cell and memory array is demonstrated. For experimental measurement, FeFET devices integrated on the 28 nm high-$\kappa$ metal gate (HKMG) technology platform are tested[19]. Figure 3a, b shows the transmission electron microscopy (TEM) and schematic

cross-section of the device, respectively. The device features an 8 nm thick doped $HfO_2$ as the ferroelectric layer and around 1 nm $SiO_2$ as the interlayer in the gate stack. The experimental setup for on-wafer characterization is shown in Fig. S1. First single cell encryption/decryption shown in Fig. 2c is demonstrated. Figure 3c, e shows the $I_D$–$V_G$ characteristics of each FeFET in a cell storing the CT of bit '0' for key bit of '1' and '0', respectively. With CT of '0', the top/bottom FeFET is programmed to the LVT/HVT, using +4V/−4V, 1$\mu$s write gate pulse, respectively. Then the decryption process simply corresponds conventional array sensing operation but with key-dependent read voltages on the two FeFETs (i.e., dashed line in Fig. 3c, e). For example, with key of '1', the top/bottom FeFETs are applied with $V_R$ (i.e., 0.6V)/0V, respectively. In this way, the top FeFET contributes a high read current, thus corresponding to the PT of bit '1'. If the key is bit '0', the read biases for the two FeFETs are swapped such that the top/bottom FeFETs receive 0V/$V_R$, respectively, where both FeFETs are cut-off, thus corresponding to the PT of bit '0'. Successful decryption can also be demonstrated for CT of bit '1' as shown in Fig. 3d, f, where the top/bottom FeFETs are programmed to the HVT/LVT state, respectively, and the same key-dependent read biases are applied. These results

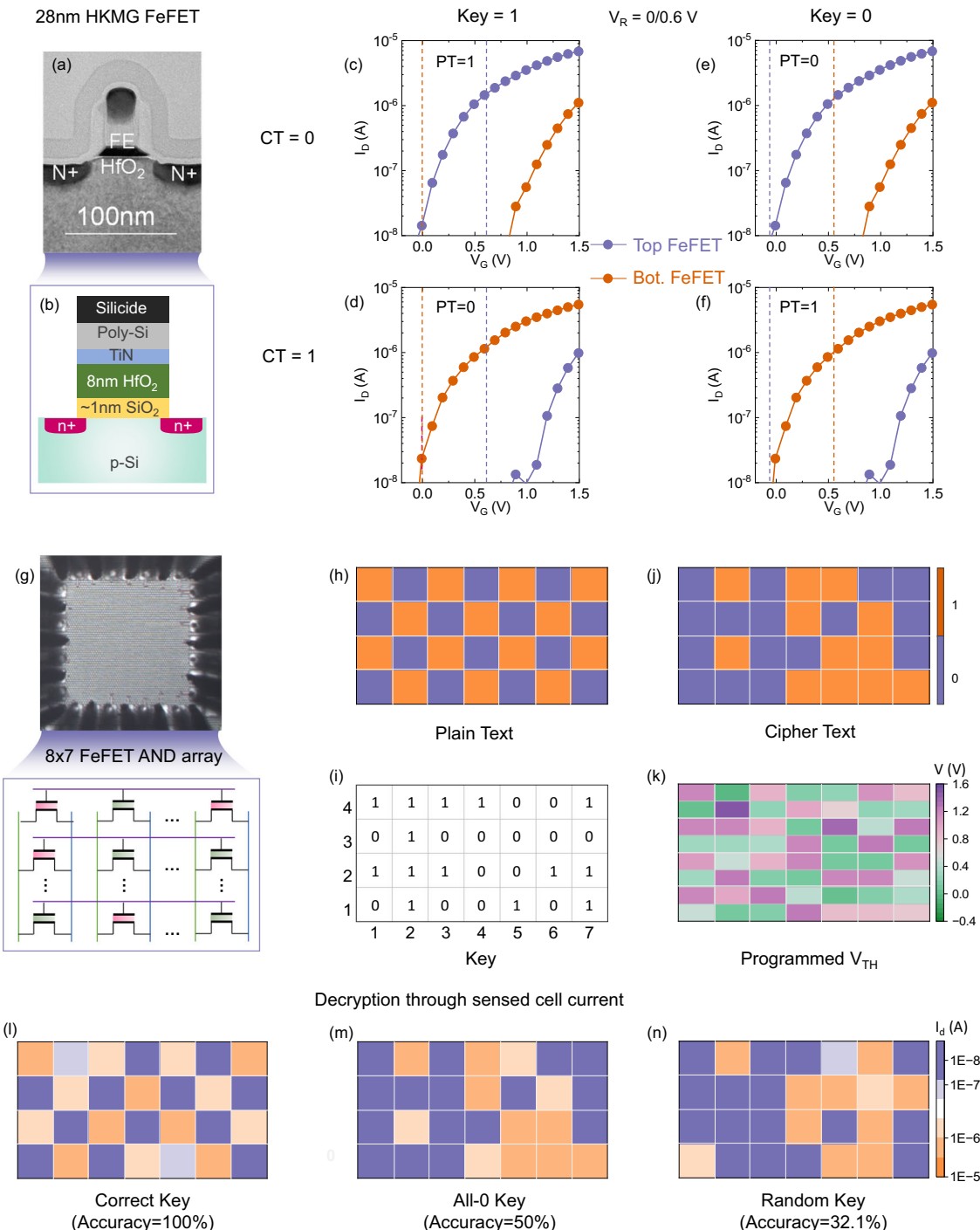

**Fig. 3 | Experimental verification. a, b** TEM and schematic cross section. **c–f** $I_D$–$V_G$ characteristics for the proposed 2FeFET memory cell. **g** The image of 8 × 7 FeFET AND array for array-level verification. **h–k** The patterns of plaintext, keys, cipher-text, and corresponding $V_{TH}$ after encryption. **l–n** In the decryption process, three conditions of applying different patterns of keys: correct keys, all-0 keys, random keys. The color bar on the right side indicates the read current measured from each cell.

demonstrate successful single-cell encryption/decryption using only in situ memory operations.

Array-level experiments and functional verification are also performed and demonstrated. Without loss of generability, FeFET AND array is adopted. Figure 3g illustrates a 8 × 7 FeFET AND memory array for measurements. Specifically, all the FeFETs have a $W/L = 0.45\,\mu m/0.45\,\mu m$. As of now, variability in FeFET has been steadily improved[20]. The array error rate has fallen below $10^{-6}$ for FeFET with $W/L = 0.2\,\mu m/$

$0.2\,\mu m$[20]. Continual material and process optimization could push the scaling of memory even further. As illustrated in Fig. 3h, a checkerboard data pattern of PT (i.e., orange boxes represent data '1'; blue boxes represent data '0'.) and random keys shown in Fig. 3i are used. To show the most general case, bit-wise encryption/decryption is validated, as encryption at a coarser granularity, i.e., row-wise or block-wise, is simply a derivation of the bit-wise case. With the PT and keys determined, the CT is simply the XOR result between the PT and

corresponding keys, as shown in Fig. 3j. Each CT bit is then stored as the complementary $V_{TH}$ states of the two FeFETs in each cell. Different write schemes along with disturb inhibition strategy can be applied[16]. In this work, block-wise erase is performed first by raising the body potential to reset the whole array to the HVT state and then selectively programming corresponding FeFETs into the LVT state. Figure 3k shows the $V_{TH}$ map of 8x7 FeFETs in the array after the encryption process, corresponding to 4x7 encrypted CT.

For the decryption process, three different scenarios are considered, i.e., using correct keys, all-0 keys, and random keys. For bit-wise encryption/decryption in AND array, since all the FeFETs in the same row share the same word line, it requires two read cycles to sense the whole row. This is because the key-dependent read voltage biases are different for key bit '1' and bit '0'. Therefore two read cycles are required where cycle 1 and 2 reads out the cells with key bit '1' and '0', respectively. Cycle 1 results are temporarily buffered and merged with cycle 2 results. Note that the additional latency can be avoided if row-wise or block-wise encryption granularity is used, where the same word line bias can be applied. As shown in Fig. 3l, under the condition of using correct keys, the user can successfully read out all PT. For attackers without the knowledge of keys, two representative scenarios are considered, where the attackers can simply apply all-0 keys or random keys. In the condition of all-0 keys, the accuracy is only 50%, as shown in Fig. 3m. With random keys, the accuracy of decryption is only 32.1%, which is much worse than other two conditions. Above all, both the functional correctness of the proposed encryption design and the resistance against attacks are verified at the cell level and array level.

## Evaluation and case study

To evaluate the feasibility and performance of the proposed in situ memory encryption /decryption scheme using FeFET memory arrays, a comprehensive evaluation is performed between this work and AES-based encryption scheme[21] in terms of area, latency, power, and throughput. For a fair comparison, an 128 × 128 FeFET AND-type array is designed in 28 nm HKMG platform and operates at 25 MHz, consistent with the reference AES work[21]. This speed serves as a pessimistic estimation of FeFET array encryption/decryption operation as it can operate at a higher speed. In addition, for memory sensing, 16 sense amplifiers (SAs) are used for illustration. If a higher sensing throughput is needed, more SAs can be deployed. For the evaluation, both the AES and proposed in situ encryption/decyrption scheme are applied. As summarized in Fig. 4, for the prior AES-based work, the area cost of its AES unit is 0.00309 mm². However, for the proposed scheme, the only functional gate required is XOR gates, whose area is negligible comparing to the whole memory area cost. Note that even though the encrypted cell size is twice of the normal FeFET cell, the area overhead of memory itself may not be 2× of normal memory area. As discussed earlier, the granularity of encrypted blocks depends on the application demands and cost budgets. Therefore, if for applications that require every FeFET cell to be encrypted, then the core array area will be twice the original unencrypted array. For certain applications, it may not be necessary to encrypt the whole memory. In that case, partial encryption can be implemented while maintaining high security. For those unencrypted blocks, normal 1T cells are adopted. Therefore, the final core area

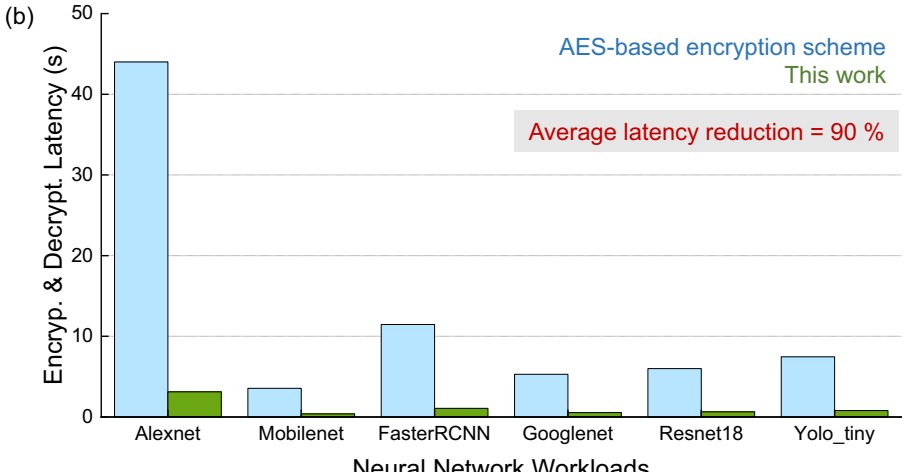

(a)

| | AES-based encryption scheme[20] | | FeFET-based encryption scheme (This work) | |
|---|---|---|---|---|
| Area (mm²) | 0.00309 (AES engine)+128x128 memory array area | | Negligible (XOR gates)+128x256 memory array area | |
| Frequency (MHz) | 25 | | 25 | |
| Power* (W) | 0.031 m | | Negligible | |
| Latency for 128-bit data (µs) | Encryption | Decryption | Encryption | Decryption |
| | 4.62 (115.5 cycles) | 4.68 (117 cycles) | 0.2 (5 cycles) | 0.64 (16 cycles[a]) |
| Throughput (Mbps) | Encryption | Decryption | Encryption | Decryption |
| | 28.32 | 28.32 | 640 | 400 |

AES-based scheme: 28 nm CMOS process. Our scheme: FeFET embedded in 28 nm HKMG platform
*: only for AES engine / XOR gate
a: Need 16 cycles (8 cycles for one pair and 8 cycles for another pair) to read 128 bits with 16 SAs. Latency can be reduced with more SAs

**Fig. 4 | Evaluation results. a** Comparison with AES-based encryption scheme[21]. **b** Latency comparison on different neural network workloads.

overhead will be 1×–2×. Moreover, the area overhead of the 2T structure only accounts for a very small part of the whole secure memory core, and is negligible compared with the area overhead of the AES engine. Besides, latency is one of the most important criteria for evaluating encryption methods. In the proposed design, the encryption and decryption latency for 128-bit data are 5 cycles and 16 cycles, respectively, which is much less than the latency penalty of the AES accelerator (115.5 cycles, 117 cycles). One thing should be noticed is that decryption latency would be reduced if more SAs are used for sensing. Moreover, at the frequency of 25 MHz, the performance of 640/400 Mbps throughput is obtained during the encryption/decryption process, which is much better than that of the AES accelerator (throughput: 28.32 Mbps). Since the power consumption of our encryption circuit is only equal to that of multiple XOR gates, it is negligible compared to the AES accelerator (0.031 mW).

In addition, to investigate the latency benefit provided by the proposed scheme compared to the conventional AES scheme when implementing data encryption and decryption with different neural network (NN) workloads, a case study is performed on 6 NN workloads which are Alexnet, Mobilenet, FasterRCNN, Googlenet, Restnet18, and Yolo_tiny via SCALE-Sim[22] which is a simulator for evaluating conventional neural network (CNN) accelerators. In this case study, we specifically consider this scenario—all the workloads are implemented into a systolic array for processing (Google TPU in this case). The encrypted weights of each neural network are pre-loaded into FeFET-based memory arrays for feeding to the systolic system after decryption. After the computation, the outputs will be read out and securely stored into the FeFET memory with encryption. As shown in Fig. 4b, the latency introduced by encryption and decryption processes of the proposed scheme is much less than that of AES-based scheme. The average latency reduction over these 6 workloads is ~90%. According to the simulation results, it shows that the proposed in situ memory encryption/decryption scheme offers significant time savings over the conventional AES scheme, especially when processing data-intensive applications, such as neural networks.

## Discussion

In summary, we propose an in situ memory encryption/decryption scheme which can guarantee high-level security by exploiting the intrinsic memory array operations while incurring negligible overheads. In addition, the functionality of the proposed scheme is verified through experiments on both device level and array level. Moreover, the evaluation results show that our scheme can hugely improve the encryption/decryption speed and throughput with negligible power cost from system-level aspect. Furthermore, an application-level case study is investigated. It shows that our scheme can achieve 90% latency reduction on average compared to the prior AES-based accelerator.

## Methods

The electrical characterization was conducted using a measurement setup comprising a PXIe System provided by NI. To access each contact of the testpad, a separate NI PXIe-4143 Source Measure Unit (SMU) was employed. Source selection for each contact was facilitated by a customized switch-matrix controlled by NI PXIe-6570 Pin Parametric Measurement Units (PPMU). The external resistor was connected to the source-terminal contact on the switch-matrix. The probe card established the connection between the switch matrix and the FeFET-structures, see Fig. S1.

## Data availability

All data that support the findings of this study are included in the article and the Supplementary Information file. These data are available from the corresponding author upon request.

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

## Acknowledgements

This work is primarily supported by the U.S. Department of Energy, Office of Science, Office of Basic Energy Sciences Energy Frontier Research Centers program under Award Number DESC0021118 (to V.N. and K.N.). The architecture part is supported by SUPREME (to K.N.) and PRISM (to V.N.) centers, two of the SRC/JUMP 2.0 centers and in part by NSF 2246149 (to S.G.) and 2212240 (to K.N.).

## Author contributions

V.N. and K.N. proposed and supervised the project. Y. Xu, Y. Xiao, Z. Zhao, X.G., and S.G. conceived the encryption/decryption schemes in different memory arrays. F.M., A.V. and T.K. performed cell and array characterization. All authors contributed to write up of the manuscript.

## Competing interests

A patent application has been submitted for this work on Sep. 27th, 2023 with the names of Y. Xu, Y. Xiao, Z. Zhao, V. Narayanan, and K. Ni on it. It has been issuing and under review by Office of Technology Management of Pennsylvania State University. The authors declare that they have no other competing interests.
