## [Peer Review File · Nature Communications]

REVIEWER COMMENTS

Reviewer #1 (Remarks to the Author):

This research paper shows an approach for securing the data stored in memory and meta data or the data being used in computational arrays. The cells are tiled into an array that are made from promising CMOS-compatible 2-transistor Ferroelectric FETs. The researchers show the proof of concept and the operation principle for the Ferroelectric FET device and a small array. The foundation pillar is XOR between a secret key and the data (in Plaintext or PT) and its encrypted version (Ciphertext or CT) during memory operation and sensing. This works with the proposed 2-transistor cell based on Ferroelectric FET device operation. They also simulate a larger array of 128 x 128 and show the benefits regarding the metrics of overhead, bandwidth, latency, and power compared with Advanced Encryption Standard (AES) implementation in the same technology node. Security is a critical concern in storing data in non-volatile memory and more so for the working data and weights in the computations of artificial intelligence workloads.

The authors did not mention more benefits that can be achieved from higher density of their 2-transistor cell compared to the SRAM and performance advantage compared to traditional known NAND and NOR based non-volatile memories.

The authors did not mention that their approach can also be augmented along and in conjunction with known AES for higher level of security if the application demands it and if the design space can tolerate higher overhead to achieve improved security. An example application is Secure Encrypted Virtualization or SEV and more ...

Researchers' data in Figure 4(a) shows that both latency and throughput are not symmetrical (or are not balanced) in their scheme/approach while the AES path shows more balanced latency and throughput for both encryption and decryption. The authors have not addressed if the application can tolerate the imbalance between encryption and decryption shown in their data in Figure 4(a) for their suggested approach. More specifically, 5-cycle latency for encryption versus 16-cycle latency for decryption. Also, 640 Mbps throughput for encryption and 400 Mbps throughput for decryption ... Design of sense amplifiers are critical in the memory operation to balance out the latency. Therefore, more design effort is required to achieve balance for the Ferroelectric FET array approach.

Paper has a typo in the pdf version in first paragraph of page 11 in line 9 where "no" needs to change to "not." Change from "they are no aware of whether the information" to "they are not aware of whether the information."

Reviewer #2 (Remarks to the Author):

In this work the authors propose a data encryption scheme to be used in FeFET AND memory arrays. Each data bit is written into two complementary cells after XOR operation using a digital key. Readout is then performed by applying read-voltages to either one of the FeFETs according to the key. This enables direct readout of the original data, leading to fast operation with only low overhead. The manuscript is well written and the main idea can be followed by the reader. However, in my eyes the work lacks a bit of novelty and is in depth discussed as detailed below.

For example, a similar concept of XOR encryption in complementary memory cells has already been discussed before for SRAM cells (for example in "Huang, Shanshi, et al. "XOR-CIM: Compute-in-memory SRAM architecture with embedded XOR encryption." Proceedings of the 39th International Conference on Computer-Aided Design. 2020."). Now the existing idea is ported into a FeFET AND array (and the other array architectures as shown in the supplementary data). However, in contrast to the SRAM cell, where the complementary data is naturally stored already, in the FeFET array for each bit a second memory cell has to be used. It means that using this algorithm in a FeFET AND array results in doubling up the required area for the memory array. Given this fact, the comparison with the other AES implementations as discussed in the paper in terms of area overhead seems not fair to me. In my eyes the overall system including memory array area and overhead for implementing the encryption/decryption scheme should be investigated.

I understand that the beauty of the concept is the little overhead that is mandatory for implementation and that the encryption/decryption is performed directly in the memory array. But that also means that the experimental data in my view boils down mainly to FeFET memory read and write operation (as it would be the case for all other memory device concepts).

Unfortunately, not too much detail is given on the array operation in this work. For example the device sizes are not disclosed (variability concern for FeFETs!), programming and inhibit schemes are not discussed and there is no discussion about their impact on the overall performance or overhead that is mandatory when realizing this concept using two complementary FeFETs. That is, there is little information that would go beyond the data discussed in the authors reference [17] and I would propose to extend this discussion with focus on the proposed concept.

Finally, there are some smaller grammar issues / typos that should be revised.

Reviewer #3 (Remarks to the Author):

The authors discuss an encryption scheme to make the stored data in a FeFET array more secure. They use a simple in-memory computing approach to generate cypered data out of the original data using two FeFETs in parallel. This is a novel an interesting idea. In detail, I have the following additional comments:

- The authors should discuss if their scheme is working for FeFET only or if it is possible to use charge trapping or floating gate cells as well and should discuss the boundary conditions for each memory cell concept in terms of programming/erase mechanism and applied voltages as well as current consumption
- The authors claim that their approach comes without overhead. However, they require a 2T cell operation to operate the memory in the secure way. Although this overhead is not comparable to the overhead of the AES encryption, they authors should discuss this drawback more clearly.
- To make the paper better understandable for the reader, the authors should add example voltages to the 2T cell drawings in fig 2c including the potential at the bulk of the device. From the reviewers perspective the current graph is hard to understand. The authors should also discuss any boundary conditions for the proposed 2T cell operation. These figures should also be added as insets to the experimental verification results (transfer curves) of fig. 3 to make those easier to understand for the reader
- The authors mention that the approach can be implemented in an AND, NAND or NOR architecture. Here they should be more precise. In fact, the AND architecture is also a NOR architecture with a separated source line per column (see [Rev_1] for example). What they call NOR in general here is more specifically a common ground NOR architecture. Moreover, in the drawing of the common ground NOR array in the supplementary the authors use a separate bitline for every column and a separate source line for every row. This is actually not the industry standard for the common ground NOR architecture were the bitline and the sourceline are always shared between two cells to minimize the amount of contacts and sourcelines. The authors should improve their nomenclature and also discuss if the scheme will work for the standard common ground NOR architecture as well. (see Fig. 12 in [Rev_1])
- Is the required 2T cell considered for the area assessment in Table 4. It seems that only the additional periphery and not the doubling of the array size was considered here?
- The authors should discuss any disturb issues that the proposed 2T operation may bring across. Is the endurance expected to be as high as for the standard memory operation?

Embedding Security into Ferroelectric FET Array via In-Situ Memory Operation

Yixin Xu¹, Yi Xiao¹, Zijian Zhao², Franz Müller³
Alptekin Vardar³, Xiao Gong⁴, Sumitha George⁵,
Thomas Kämpfe³, Vijaykrishnan Narayanan¹, Kai Ni^{2*}

¹Pennsylvania State University, State College, PA 16802, USA

²Rochester Institute of Technology, Rochester, NY 14623, USA

³Fraunhofer IPMS, Dresden, Germany

⁴National University of Singapore, Singapore

⁵North Dakota State University, Fargo, ND 58102, USA;

*To whom correspondence should be addressed

Email: kai.ni@rit.edu

The authors would like to thank the reviewers and editors for going over our revised manuscripts.

For the additional concern, we have modified our manuscript and provided a point-to-point response here.

Review question 1.1

The authors did not mention more benefits that can be achieved from higher density of their 2-transistor cell compared to the SRAM and performance advantage compared to traditional known NAND and NOR based non-volatile memories.

Thanks for the reviewer bringing this up. We can add that into the text. It is true that the our proposed solution, though with two transistors in a single cell, still shows density improvement over the SRAM counterpart. In the original manuscript, we have been focusing on the security

concerns in non-volatile memories, thereby only mentioning the conventional encryption method (i.e., AES) for non-volatile memories.

Bearing the similar single transistor structure, the conventional NAND and NOR flash memories can also be encrypted/decrypted with the proposed techniques. However, flash generally require a large operation voltages and a long write latency, therefore exhibiting a poor performance compared with FeFET, as pointed out by the reviewer.

Relevant texts are added into first paragraph on page 9 and page 12.

Review question 1.2

The authors did not mention that their approach can also be augmented along and in conjunction with known AES for higher level of security if the application demands it and if the design space can tolerate higher overhead to achieve improved security. An example application is Secure Encrypted Virtualization or SEV and more...

Thanks for the reviewer's suggestions. Yes, our encryption strategy can definitely work with AES together in order to provide higher security for some specific applications. For example, we can pick the AES as the first cipher and our work as the second cipher. During encryption, the plaintexts can firstly send to the AES engine to get the ciphertexts which would be sent as inputs of our XOR cipher to do the second encryption. The ciphertexts after these 2 ciphers can finally be stored in the FeFET arrays with improved security. Similarly, for decryption, the data in the

memory is read out using our decryption method then sent to AES to obtain the actual plaintexts.

Relevant texts are added into the last paragraph on page 8.

Review question 1.3

Researchers' data in Figure 4(a) shows that both latency and throughput are not symmetrical (or are not balanced) in their scheme/approach while the AES path shows more balanced latency and throughput for both encryption and decryption. The authors have not addressed if the application can tolerate the imbalance between encryption and decryption shown in their data in Figure 4(a) for their suggested approach. More specifically, 5-cycle latency for encryption versus 16-cycle latency for decryption. Also, 640 Mbps throughput for encryption and 400 Mbps throughput for decryption ... Design of sense amplifiers are critical in the memory operation to balance out the latency. Therefore, more design effort is required to achieve balance for the Ferroelectric FET array approach.

Thank the reviewer for raising this point. In our opinion, it is not absolutely necessary to have symmetric encryption and decryption throughput. But if for some applications that require symmetric encryption and decryption latency/throughput, our design can definitely meet that requirement. This is because the encryption throughput depends on the FeFET programming speed and the decryption throughput depends on the FeFET sensing and the parallelism of the sensing circuitry. The decryption latency/throughput shown in Fig.4(a) can be improved because it highly depends on the number of sense amplifiers (In Fig.4(a), we use 16 sense amplifiers to showcase).

If we increase the number of sense amplifiers with proper design, the latency/throughput for encryption and decryption can be balanced, as we mentioned in the first paragraph on page 19. It should be observed that both encryption and decryption latency of our design are much lower than those of AES design.

Relevant texts are highlighted in the first paragraph on page 18.

Review question 1.4

Paper has a typo in the pdf version in first paragraph of page 11 in line 9 where “no” needs to change to “not.” Change from “they are no aware of whether the information” to “they are not aware of whether the information.”

Thank you for pointing this typo out. We’ve fixed it in the revised manuscript.

Review question 2.1

For example, a similar concept of XOR encryption in complementary memory cells has already been discussed before for SRAM cells (for example in “Huang, Shanshi, et al. ”XOR-CIM: Compute-in-memory SRAM architecture with embedded XOR encryption.” Proceedings of the 39th International Conference on Computer-Aided Design. 2020.”). Now the existing idea is ported into a FeFET AND array (and the other array architectures as shown in the supplementary data). However, in contrast to the SRAM cell, where the complementary data is naturally stored

already, in the FeFET array for each bit a second memory cell has to be used. It means that using this algorithm in a FeFET AND array results in doubling up the required area for the memory array. Given this fact, the comparison with the other AES implementations as discussed in the paper in terms of area overhead seems not fair to me. In my eyes the overall system including memory array area and overhead for implementing the encryption/decryption scheme should be investigated.

Thanks for the reviewer for providing this reference. After carefully reading through the reference paper¹, we think that our novelty and contribution still stand and our work is significantly different from that SRAM-CIM work on the following aspects although both of us are based on XOR ciphers.

First of all, their design basically leverages the normal SRAM cells but with two separate WL ports. The decryption is conducted by choosing the appropriate side for output to realize the XOR-cipher function. Therefore, although SRAM can naturally store complementary data and the structure is compact, this design destroys the original symmetry of the SRAM structure (i.e., this allows differential sensing) and changes to a single-ended design, making it incompatible with normal SRAM arrays. In contrast, our proposed design is fully compatible with the normal memory array.

Secondly, the authors of the SRAM-CIM paper mentioned that there would be read currents on both directions – charge current and discharge current, which not only makes their design, i.e., CIM results, more sensitive to mismatch in the two current paths but also cause higher energy consumption. That's a very critical problem. However, in our design, we don't have this kind of

issue because the read current in our design only passes one direction.

Thirdly, the peripheral circuitry of the SRAM-CIM design is more complex than ours because they need two ADCs for each BL and \overline{BL} pair and additional circuit used for summing up converted analog results. Although the authors argued that in some cases they can use lower-resolution ADCs with smaller area because each side might see a smaller partial sum than the original case in some cases, they still need to consider the worst case scenario in which their design still need a pair of ADCs for each BL and \overline{BL} pair. Thus, they cannot mitigate the extra cost from ADCs. However, our design only require one ADC per column since we only have one BL current need to be sensed.

Besides, as claimed in the SRAM-CIM paper, their design requires one reference array with the same size as the normal CIM array to generate reference signals for ADCs. In contrast, our design doesn't need that.

Relevant texts are added into first paragraph on page 12.

Review question 2.2

I understand that the beauty of the concept is the little overhead that is mandatory for implementation and that the encryption/decryption is performed directly in the memory array. But that also means that the experimental data in my it boils down mainly to FeFET memory read and write operation (as it would be the case for all other memory device concepts).

Unfortunately, not too much detail is given on the array operation in this work. For example the device sizes are not disclosed (variability concern for FeFETs!), programming and inhibit schemes are not discussed and there is no discussion about their impact on the overall performance or overhead that is mandatory when realizing this concept using two complementary FeFETs. That is, there is little information that would go beyond the data discussed in the authors reference [17] and I would propose to extend this discussion with focus on the proposed concept.

Thanks for raising these points. We focused on demonstrating the concepts in the earlier version and did not add these details. They have been added in the revised manuscript. Specifically, all the FeFETs have a $W/L=0.45\mu\text{m}/0.45\mu\text{m}$. As of now, variability in FeFET has been steadily improved². The array error rate has fallen below 10^{-6} for FeFET with $W/L=0.2\mu\text{m}/0.2\mu\text{m}$ ². It is believed that with further material and process optimization, further scaling of memory is possible.

Figure R1: Three steps for programming of the FeFET AND array to implement the encryption scheme.

Regarding the programming scheme, as shown in Fig. R1, we firstly program the whole block which need to be encrypted to HVT state by asserting all WLs at $-V_w$ (i.e., $V_w=3.3\text{ V}$

in this work), then program one FeFET in each 2FeFET cell to the LVT state by applying $+V_W$ on the corresponding WL. Therefore a total of 3 cycles are required to implement the encryption scheme. For proper operation of the array, inhibition bias schemes need to be applied to prevent undesired programming to unselected cells. Two schemes are generally available, i.e., $V_W/2$ and $V_W/3$ scheme³. Here, we choose $V_W/3$ scheme to minimize the disturb, as shown in Fig. R1.

Discussion on FeFET variability is added on page 14. Discussion on programming and inhibit schemes is added in the supplementary. Fig. R1 is added as Fig. S4 in the supplementary. New references are added as Reference 18 and 22 in the manuscript.

Review question 3.1

The authors should discuss if their scheme is working for FeFET only or if it is possible to use charge trapping or floating gate cells as well and should discuss the boundary conditions for each memory cell concept in terms of programming/erase mechanism and applied voltages as well as current consumption

Yes, in principle our schemes can be also applied with charge trapping and floating gate cells. The programming/erase mechanism of flash arrays should be similar but with a few minor differences because of different electrical characteristics. For both FeFET arrays and flash arrays, we can use the same 'erase-program' mechanism for encryption – firstly erase the whole block and then selectively program cells. However, flash generally needs a larger write voltage (e.g., ± 10 V) and a slower write speed due to its thick gate stack and poor efficiency for memory write. HfO₂

based FeFET, however, can be written by below 4V and 10 ns write pulses, demonstrating superior write performance. Demonstration of FeFET on advanced transistor technologies, such as the FinFET⁴ and gate-all-around transistor^{5,6}, have been reported. Therefore, charge trapping/floating gate cells can, in principle, work in our design, but with more challenges.

Relevant texts are added into the second paragraph on page 12.

Review question 3.2

The authors claim that their approach comes without overhead. However, they require a 2T cell operation to operate the memory in the secure way. Although this overhead is not comparable to the overhead of the AES encryption, they authors should discuss this drawback more clearly.

Thanks for bringing this point out. It is true that at the single cell level, the encrypted cell size is twice the normal FeFET cell. Therefore, if for applications that require every FeFET cell to be encrypted, then the core array area will be twice the original unencrypted array. That is the overhead of the proposed design. As we also mentioned in page 11, the granularity of encrypted blocks depends on the application demands and cost budgets. For certain applications, it may not be necessary to encrypt the whole memory. In that case, we can implement partial encryption while maintaining high security. For those unencrypted blocks, we can still implement 1T cells. Therefore, the final core area will be $1\times - 2\times$ of the original unencrypted FeFET core array. We also estimate the areas of the array cores with AES/our scheme, respectively. Since both of the works use 28 nm technology, we can roughly calculate the area of the AES core as follows – 0.00309

mm^2 (AES engine) + $128*128*6*28nm*28nm$ (128*128 memory array) = $0.003167 mm^2$, and similarly, the area of our FeFET secure core is – negligible (XOR gates) + $128*256*6*28nm*28nm$ = $0.0001541 mm^2$. As the estimation shows, the area overhead of our 2T structure only accounts for a very small part, and is not comparable to the area overhead of the AES engine.

Relevant texts are added on page 18 in the section of evaluation and case study.

Review question 3.3

To make the paper better understandable for the reader, the authors should add example voltages to the 2T cell drawings in fig 2c including the potential at the bulk of the device. From the reviewers perspective the current graph is hard to understand. The authors should also discuss any boundary conditions for the proposed 2T cell operation. These figures should also be added as insets to the experimental verification results (transfer curves) of fig. 3 to make those easier to understand for the reader.

Thanks for the recommendation. We have added all the missing details in the manuscript. Fig. R1 shows the three cycle operation to implement the encryption. Programming voltages on each line and body, along with the inhibition bias schemes, are illustrated. The decryption voltages (i.e., read voltages) are chosen as 0V and $V_R=0.6V$, respectively as shown in Fig.3 in the main manuscript.

Fig. 2 and Fig. 3 have been updated with example voltages. Discussion on boundary condi-

tions is added in the supplementary.

Review question 3.4

The authors mention that the approach can be implemented in an AND, NAND or NOR architecture. Here they should be more precise. In fact, the AND architecture is also a NOR architecture with a separated source line per column (see [Rev1] for example). What they call NOR in general here is more specifically a common ground NOR architecture. Moreover, in the drawing of the common ground NOR array in the supplementary the authors use a separate bitline for every column and a separate source line for every row. This is actually not the industry standard for the common ground NOR architecture where the bitline and the sourceline are always shared between two cells to minimize the amount of contacts and sourcelines. The authors should improve their nomenclature and also discuss if the scheme will work for the standard common ground NOR architecture as well. (see Figure 12 in [Rev1]).

Thanks for the reviewer for pointing this out. We redraw the figure in supplementary, as shown in Fig. R2(b). Also, we put the common NOR array structure in Fig. R2(a) as a reference. We use purple/green/yellow lines to indicate WL/BL/SL in both two arrays. As can be seen, all the FeFETs in one column share the BL, and every neighbor FeFET share their SLs together. Since the SL remains at 0V in both cases, the encryption/decryption schemes stay unchanged and work for this NOR array structure.

Figure R2: (a) A common ground NOR array. (b) Our NOR array.

Fig. S3 has been updated in the supplementary.

Review question 3.5

Is the required 2T cell considered for the area assessment in Table 4. It seems that only the additional periphery and not the doubling of the array size was considered here?

Yes. In Table 4(a), we only compare their cipher part in term of area cost, i.e., the AES engine and the XOR gates. As for the memory array size, we didn't include it in our comparison as the actual FeFET array size varies depending on different cost and security demands, not simply doubling the normal array size. To eliminate the possible misunderstanding, we've updated the area part to include the memory area in Table 4(a). Besides, we also estimate the areas of the array cores with AES/our scheme, respectively. Since both of the works use 28 nm technology, we can roughly calculate the area of the AES core as follows – 0.00309 mm^2 (AES engine) + $128 \times 128 \times 6 \times 28 \text{ nm} \times 28 \text{ nm}$ (128×128 memory array) = 0.003167 mm^2 , and similarly, the area of

our FeFET secure core is – negligible (XOR gates) + $128*256*6*28\text{nm}*28\text{nm} = 0.0001541 \text{ mm}^2$.

As the estimation shows, the area overhead of our 2T structure only accounts for a very small part, and is not comparable to the area overhead of the AES engine.

Table. 4 has been updated with the area information. Discussion on the area cost of 2T cells is added into page 18.

Review question 3.6

The authors should discuss any disturb issues that the proposed 2T operation may bring across. Is the endurance expected to be as high as for the standard memory operation?

The proposed encryption/decryption scheme is fully compatible with the original memory array, introducing no additional reliability concerns. As shown in Fig.R1, compared with unencrypted cells, which occupied only one row, the encrypted cells need one additional programming cycle. The inhibition schemes are exactly the same as the original array. No new disturb cases are introduced. Endurance will be impacted to the extent that an extra write results to an additional complementary row to achieve the encryption. However, the mechanisms for endurance do not change.

Relevant texts are added in the supplementary.

References

1. Huang, S., Jiang, H., Peng, X., Li, W. & Yu, S. Xor-cim: Compute-in-memory sram architecture with embedded xor encryption. In *2020 IEEE/ACM International Conference On Computer Aided Design (ICCAD)*, 1–6 (2020).
2. Beyer, S. *et al.* Fefet: A versatile cmos compatible device with game-changing potential. In *2020 IEEE International Memory Workshop (IMW)*, 1–4 (IEEE, 2020).
3. Ni, K., Li, X., Smith, J. A., Jerry, M. & Datta, S. Write disturb in ferroelectric fets and its implication for 1t-fefet and memory arrays. *IEEE Electron Device Letters* **39**, 1656–1659 (2018).
4. Yan, S.-C. *et al.* High speed and large memory window ferroelectric hfzro finfet for high-density nonvolatile memory. *IEEE Electron Device Letters* **42**, 1307–1310 (2021).
5. Huang, W. *et al.* Ferroelectric vertical gate-all-around field-effect-transistors with high speed, high density, and large memory window. *IEEE Electron Device Letters* **43**, 25–28 (2021).
6. Lee, S.-Y., Lee, C.-C., Kuo, Y.-S., Li, S.-W. & Chao, T.-S. Ultrathin sub-5-nm hf zro for a stacked gate-all-around nanowire ferroelectric fet with internal metal gate. *IEEE Journal of the Electron Devices Society* **9**, 236–241 (2021).

REVIEWERS' COMMENTS

Reviewer #1 (Remarks to the Author):

This research paper shows an approach for securing the data stored in memory and meta data or the data being used in computational arrays when using ferroelectric-based non-volatile memories, particularly a CMOS-compatible 2-transistor Ferroelectric FETs as they are being researched. The foundational pillar is based on XORing a secret key and the data (in Plaintext) and its encrypted version (in Ciphertext) during memory operation and sensing. The technology and proof of concept that the authors are researching enables them to achieve this goal efficiently as measured by the metrics listed in their paper. The authors have answered my questions and comments during the paper review process and have changed the manuscript accordingly.

Reviewer #2 (Remarks to the Author):

I thank the authors for their answers that addressed my concerns.

The revised manuscript might be published in its present form.

There is just a minor aspect that I would recommend authors to address in a minor revision:

Now with the proposed writing scheme that requires an erase to high- V_{th} state before each program to low- V_{th} state complementary values have to be written and a single-bit alteration is not possible anymore - only row-wise writing will be possible. Authors should briefly mention this issue and discuss whether there is any impact on the array and application cases.

Reviewer #3 (Remarks to the Author):

The authors have sufficiently answered all my questions. I therefore recommend to publish the paper in its current form.

Embedding Security into Ferroelectric FET Array via In-Situ Memory Operation

Yixin Xu¹, Yi Xiao¹, Zijian Zhao², Franz Müller³
Alptekin Vardar³, Xiao Gong⁴, Sumitha George⁵,
Thomas Kämpfe³, Vijaykrishnan Narayanan¹, Kai Ni^{2*}

¹Pennsylvania State University, State College, PA 16802, USA

²University of Notre Dame, Notre Dame, IN 46556, USA

³Fraunhofer IPMS, Dresden, Germany

⁴National University of Singapore, Singapore

⁵North Dakota State University, Fargo, ND 58102, USA;

*To whom correspondence should be addressed

Email: kni@nd.edu

Review question 2.1

There is just a minor aspect that I would recommend authors to address in a minor revision: Now with the proposed writing scheme that requires an erase to high-V_{th} state before each program to low-V_{th} state complementary values have to be written and a single-bit alteration is not possible anymore - only row-wise writing will be possible. Authors should briefly mention this issue and discuss whether there is any impact on the array and application cases.

Thanks for pointing this out. In this work, we choose to use the erase-program scheme with the fixed body bias (Fig. R1(a)) so that fails to support single-bit alterations. However, if we implement column-wise body contacts, we can selectively program one single bit^{1,2} but with the extra cost of separate body contacts (Fig. R1(b)).

Array with one shared body contact

Array with column-wise shared body contacts

Figure R1: (a) The AND array with one shared body contact. (b) The AND array with column-wise body contacts.

Relevant texts are added into first paragraph on page 9. New reference has been cited as reference 17 in the manuscript.

References

1. Jiang, Z. *et al.* On the feasibility of 1t ferroelectric fet memory array. *IEEE Transactions on Electron Devices* **69**, 6722–6730 (2022).
2. Xiao, Y. *et al.* On the write schemes and efficiency of fefet 1t nor array for embedded nonvolatile memory and beyond. In *2022 International Electron Devices Meeting (IEDM)*, 13.6.1–13.6.4 (2022).